# Real-Time PCR-Based Methods for Detection of Hepatitis E Virus in Pork Products: A Critical Review

**DOI:** 10.3390/microorganisms10020428

**Published:** 2022-02-12

**Authors:** Nigel Cook, Martin D’Agostino, Ann Wood, Linda Scobie

**Affiliations:** 1Jorvik Food Safety Services, York YO32 2GN, UK; 2Campden BRI, Chipping Campden GL55 6LD, UK; martin.dagostino@campdenbri.co.uk (M.D.); ann.wood@campdenbri.co.uk (A.W.); 3Department of Biological and Biomedical Sciences, Glasgow Caledonian University, Glasgow G4 0BA, UK

**Keywords:** hepatitis E virus, detection, real-time RTPCR, pork products, sample treatment

## Abstract

Standard methods for detection of hepatitis A virus and norovirus in at-risk foodstuffs are available, but currently there is no standard method for detection of hepatitis E virus (HEV) in pork products or other foods that can be contaminated with the virus. Detection assays for HEV are mainly based on nucleic acid amplification, particularly the reverse transcription polymerase chain reaction (RTPCR) in real-time format. RTPCR-based methods can be sensitive and specific, but they require a suite of controls to verify that they have performed correctly. There have been several RTPCR methods developed to detect HEV in pork products, varying in details of sample preparation and RTPCR target sequences. This review critically discusses published HEV detection methods, with emphasis on those that have been successfully used in subsequent studies and surveys. RTPCR assays have been used both qualitatively and quantitatively, although in the latter case the data acquired are only reliable if appropriate assay calibration has been performed. One particular RTPCR assay appears to be ideal for incorporation in a standard method, as it has been demonstrated to be highly specific and sensitive, and an appropriate control and calibration standard is available. The review focuses on the detection of HEV in pork products and similar foodstuffs (e.g., boar). The information may be useful to inform standardisation activities.

## 1. Introduction

The emergence of hepatitis E virus (HEV) as a zoonotic pathogen which may be transmitted by foods, especially through the pork supply chain, has resulted in a great deal of research being carried out by food and veterinary virologists across the globe to establish where links exist and how we may quantify the risks posed by the virus, and in developing methods to detect the virus in our foods [1]. In equal measure, there has been much concern from the food industry, especially the pork supply industry, as sporadic outbreaks of hepatitis E that are not travel related have been epidemiologically linked to food items. Ad hoc testing of retail pork-based products in various countries has resulted in a flurry of media activity, often negative in nature, with consequences that have had an impact on political and economic exportation policies [2].

Among the reasons why there is such concern about HEV are the seemingly endemic levels of the zoonotic genotype 3 (gt3) strain in circulation in European pigs [3], its detection in foods that do not include a heat processing step [1], and the lack of available methods to demonstrate infectivity of the virus and its infectious dose in humans [4,5]. Some products may contain raw ingredients, which are at a higher risk of containing the virus. Increased screening of blood donors for HEV antibodies has demonstrated that in many cases there is a percentage of the population which has been exposed to the virus [6,7]. This includes those who have not travelled and who may, or may not, have displayed any symptoms of hepatitis infection at some point in their lives, providing further evidence that items in their diet may have exposed them to the infection. A lack of clear answers to these and many other questions hampers efforts to determine the control measures that should be put in place during food production to ensure infectivity is reduced or eliminated. Often the only solution appears to be elimination of the virus from the pig herds that harbour the virus; however, the reasons why this virus has established itself in these environments is unclear, as is how it was introduced in the first place. Without this knowledge, elimination is difficult. 

Given these significant challenges, which are not easily resolved, laboratories have developed a range of in-house detection methodologies. Some methods have been based on cell culture systems. Not only do these methods detect the virus and provide detail on infectivity, but most are rapid methods based on nucleic acid amplification (NAA), to attempt to directly detect the virus in foods of concern at a reasonable cost. However, this approach has suffered from an absence of standardisation, which in turn results in a lack of reproducibility between laboratories and a deficiency when it comes to comparability of results between laboratories. In order to rectify this, a standardised method is required to allow laboratories involved in testing foods for the presence of HEV. This will allow both regulators and the industry to better determine risk factors, and have more confidence in outputs of surveillance studies and ultimately in creating control strategies. A similar approach was taken some years previously with norovirus and hepatitis A detection in fresh and frozen produce, which has resulted in the publication of ISO methods that can be used as globally accepted standard methods [8,9]. The International Standards Organisation has recently created a working group (ISO/TC34/SC9/WG31) to develop a standard for the detection of HEV in foods, with a potential focus on pork products; the standard will be based on nucleic acid amplification targeting HEV RNA sequences. This review focuses on the detection of HEV in pork products and similar foodstuffs using RTPCR and the factors that need to be taken into consideration when evaluating detection data.

## 2. Reverse Transcription Polymerase Chain Amplification-Based Detection of Foodborne Viruses: An Overview of Basic Principles and Controls

The NAA assay, which has been predominantly employed in analysis of HEV in pork products, is the reverse transcription polymerase chain reaction (RTPCR). Viral RNA is transcribed by the enzyme reverse transcriptase into cDNA, containing a strand of complementary DNA linked to the original strand of RNA. The cDNA is then cycled through the PCR process, resulting in massive amplification of double-stranded DNA fragments or amplicons, containing nucleotide sequences mirroring those of the original viral target.

In the early use of PCR, amplicons were visualised by gel electrophoresis of the completed reaction solution, followed by staining with a fluorescent dye; amplicons were determined to be specific for the target sequence by their size. This approach is termed “conventional PCR” or “conventional RTPCR” in this review. Conventional PCR has been superseded by so called “real-time” PCR (subsequently termed qPCR in this review); a largely automated process in which the formation of amplicons is monitored as the reaction proceeds (although in actual practice the signal is viewed when the reaction has completed). RTPCR can be performed qualitatively or quantitatively. In a qualitative format, the reaction gives a “presence or absence” result. In a quantitative format, the reaction is calibrated using a suspension of pre-quantified targets, diluted to reflect the range of potential target quantities that may be present in a sample. Detection of RNA viruses such as HEV requires that the calibration standards be composed of RNA, so that the RT step is taken into account; however, several assays have used DNA plasmid constructs, which reflect only the PCR process. In this report, RTPCR is used for non-quantitative assays, RTqPCR for assays that have been quantified using DNA, and qRTPCR for assays that have been quantified using RNA standards. 

Methods used to detect viruses in foods are complex, involving several basic parts requiring several individual steps in each, and a suite of controls is necessary to ensure the reliability of results [10]. To extract viruses from a complex food sample, in which they may be present in low numbers, a multistep process must be employed, and the possibility of failing to extract viruses, or extracting them with low efficiency, increases with the complexity of the matrix and the number of steps in the sample treatment. In an ideal method, a sample process control (SPC) is included to verify that there have been no issues with sample treatment. The SPC is a non-target virus, added immediately upon sample receipt in the laboratory [11]; if it is detected with acceptable recovery efficiency the method is considered to have been effective for that sample. 

Complex food matrices, such as pork products, can contain substances inhibitory to RTPCR, and if they are co-extracted with the target, reaction failure may result. To control for this possibility, RTPCR assays generally include an amplification control (AC). This is an RNA oligonucleotide comprising sequences from the virus target; if it is detected with acceptable efficiency the RTPCR is considered to have been effective for that sample. The most commonly used AC type is the internal amplification control (IAC) [12], which is added to the RTPCR mix prior to amplification and co-amplified with the target. 

An RTPCR signal by itself cannot indicate whether the detected virus is infectious, as RNA can persist in non-infectious particles [13]. Various adaptations have been attempted to allow RTPCR to mediate detection of infectious, or at least intact and therefore potentially infectious, particles. Treatment of virus suspensions with RNase to destroy exposed RNA can selectively detect intact virus particles [14]; similarly, pre-treatment with RNA-intercalating dyes can prevent RTPCR from amplifying nucleic acid sequences from damaged viruses [15]. Neither of these adaptations, however, can selectively detect infectious virus. They can only indicate a potential for infectivity; however, their use could provide more information than an unadapted RTPCR [16]. 

The following review critically discusses the published RTPCR-based methods for the detection of HEV in foods, discussing the sample treatments employed, and any incorporation of controls. The information may be useful to inform standardisation activities.

## 3. Real-Time RTPCR-Based Detection Methods for HEV in Pork Products

There has been debate as to whether detection for HEV should be quantitative or qualitative and, as seen for the extraction procedures, a number of molecular methods have also been published for detection. Methods vary from detection of specific genotypes to pan-detection methods, and there is also variation in the controls used to validate the assays. Initially, most assays were designed to detect HEV in clinical samples and have been then taken forward for detection in food. This section focuses on PCR-based methods observed in the literature as this is the most likely approach. 

While investigating the sero-prevalence of HEV in the south-west of France, Mansuy et al. (2004) [17] developed an assay, using the TaqMan format, with primers targeted against open reading frame (ORF)2 sequences. Although not stated directly, the assay was apparently universal for HEV. The assay was used to detect HEV in patients’ serum samples; subsequent sequencing was used to characterise the virus as gt3. Although no information on calibration was given, the limit of detection (LOD)was reported as an estimated 1 × 10^3^ copies mL^−1^ serum. Similarly, in 2006, three publications described similar assays; a universal HEV assay using the same format as above and targeting ORF 2 sequences was developed by Enouf et al. [18] and Ahn et al. [19], and by Jothikumar et al. [20], who developed a universal RTPCR assay, targeting ORF3 sequences for clinical analysis, and quantitation was performed with DNA using a plasmid construct. 

Quantitation using RNA calibration standards was performed by Ahn et al. [19] where HEV ORF2 sequences were cloned into a plasmid, which was then transcribed in vitro to produce cDNA with a detection limit of 168 genome equivalents (GE; this term is used to take into account the possibility that RNA fragments containing the primer sequences can be detected, and not always whole genomes). 

The main difference here was that the assay by Jothikumar et al. [20] demonstrated specificity using a panel of non-HEV strains. None of these four assays included an AC.

These four assays were subsequently evaluated by Ward et al. [21], using a panel of pig faecal and serum samples. The assay of Jothikumar et al. [20] was found to be the most effective, detecting HEV in more samples than the other assays, and with greater sensitivity as judged by lower Ct values, suggesting the potential for this assay to be used in HEV detection in food.

For improved detection and to include a SPC, Ward et al. [21] merged the Jothikumar et al. (2006) assay with an in-house assay for feline calicivirus (FCV) to produce a multiplex RTPCR. They used feline calicivirus (FCV) as both a SPC and a heterologous IAC, adding virus particles to samples prior to NAA extraction. Plasmid standards were used to calibrate the multiplex assay. This assay was subsequently used to detect HEV in organs (including liver and muscle) and tissues of pigs at slaughter [22]. No HEV was detected in muscle samples, but the virus was detected in liver. Quantitative data was obtained. 

To further determine the usefulness of PCR in HEV detection in food, in a survey of pork chops and pork livers sold at retail outlets in Canada, Wilhelm et al. [23] used the assay of Ward et al. [21] following the application of separate sample preparation procedures for each food type. For liver, 312 mg portions were added to a commercial lysis buffer and homogenised with ceramic beads. After centrifugation, 4 mL supernatant was used for NA extraction by commercial kit. The procedure for pork chops was similar except that after centrifugation, proteinase K digestion was performed, followed by an additional round of centrifugation prior to NA extraction. Approximately 57% of livers analysed had HEV nucleic acid detected, with no chops testing HEV-positive. Wilhelm et al. [24], revisiting the data from [23], considered that the likelihood of HEV detection was greater with liver than with pork chops, probably due to the latter being a more difficult matrix to process.

At a similar time, the assay of Kaba et al. [25], targeting ORF2 sequences, was developed to detect transmission in piglets and was used by Colson et al. [26] to detect HEV in figatellu, a sausage made from raw pig liver. In contrast to the study by Wilhelm et al. [23] the assay was not quantitative and no AC or SPC was included.

Detection of HEV in animal products became more prevalent in the literature with the main aim to detect HEV gt3, most commonly found in pork products. In addition, assays were improved to include relevant controls to validate the data.

HEV was detected in samples of pig liver sold at retail outlets in Germany by Wenzel et al. [27], using the RTqPCR assay described by Jothikumar et al. [20] using the human coxsackievirus B added to the sample prior to homogenisation as SPC, but did not include an AC control. An additional sample treatment was employed by Wenzel et al. [27] to include the testing of a capsid integrity-based RTqPCR approach to determine potential virus infectivity and as a comparison to straightforward NAA detection as previously described [28]. The detections of HEV in the two studies were broadly in accordance, but more evaluation of this approach is necessary before it can be used with confidence for suggesting infectivity; meanwhile, the caveat discussed above regarding infectivity assays remains pertinent. 

Diez-Valcarce et al. [29] proposed the use of murine norovirus (MNV) as a suitable SPC for analysis of foods for enteric viruses, and it was incorporated in the fully controlled method for HEV detection in pork liver and muscle samples; this method was used in the studies of Di Bartolo [30] and Berto et al. [31] both of which used the sample treatment procedure of Bouwknegt et al. [32]. In each study, the assay of Jothikumar et al. [20] was utilized, and was modified by the use of an RNA IAC, as designed by Diez-Valcarce et al. [33]. No quantitation was performed in either study. Subsequent studies by Di Bartolo et al. [34] and De Sabato et al. [35] used the same method, but the assay was calibrated using plasmid-transcribed RNA sequences identical to those of the HEV target sequences to produce fully quantitative results. Garcia et al. [36] modified the sample treatment procedure by incorporation. of a commercial phenol:chlororm based reagent prior to chloroform extraction; they calculated the extraction efficiency of this procedure as being approximately 50%.

A triplex qRTPCR assay, incorporating the primer sets of Gyarmati et al. [37] and Jothikumar et al. [20], and a heterologous IAC, was used as the basis of a method to detect HEV in wild boar and mouflon [38]. Sample treatment was similar to that of Bouwknegt et al. [32]. RNA standards were used for quantification, and the LOD of the assay was determined to be 50 GE. Son et al. [39] also used the assay of Jothikumar et al. [20] for HEV detection in pig liver and quantified the data using RNA standards. No SPC or AC was reported. No LOD was reported.

A survey of foods containing raw pork liver was carried out in France by Pavio et al. [40]. The foods included figatellu, dried liver, dried and fresh sausages, and liver paste (quenelle), and HEV was detected in samples of each type. No SPC was used and, again, the assay of Jothikumar et al. [20] was employed, as modified by Barnaud et al. [41] to include an EAC consisting of HEV RNA. Quantitative data were reported, and as the assay was calibrated using HEV RNA, these can be regarded as accurate. The assay of Barnaud et al. [41] was subsequently used in a survey of pig liver and meat obtained at French slaughterhouses [42], and to detect HEV in the muscle juice of experimentally infected pigs [43]. 

Comparing nine in-house sample treatment procedures for detection of HEV in pig liver sausages, Martin-Latil et al. [44] selected as optimal a process based on the use of polyethylene glycol (PEG) as a virus particle flocculant. Artificially contaminated food samples were used as test materials. MNV was used as SPC. qRTPCR was performed using the duplex HEV/MNV assay of Martin-Latil et al. [45] using HEV RNA calibration standards, MNV acting as both SPC and a heterologous IAC. The sample treatment procedure of Martin-Latil et al. [44] was employed to facilitate quantitative detection of HEV in liver samples using a subsequent digital RTPCR assay (Martin-Latil et al. [46]. 

An evaluation of virus extraction procedures was performed by Hennechart-Collette et al. [47]. Six procedures involving permutations of sample size, homogenisation liquid (distilled water; dH_2_O) volume, and homogenisation techniques were examined, using pork liver, sausage and figatellu that had previously [44] been found to be HEV-contaminated. The qRTPCR of Jothikumar et al. [20], quantified using RNA standards, was used to evaluate the effectiveness of each procedure by comparing the GE copy numbers obtained. 

Szabo et al. [48] reported that a sample treatment based on the use of a commercial phenol:chloroform-based reagent gave a better recovery (4.90%) of HEV in artificially contaminated pork products than the procedures of Colson et al. [26], Di Bartolo et al. [30] and Martin-Latil et al. [44]. The RTPCR assay of Jothikumar et al. [20], with quantitation using RNA standards, was employed to detect viral RNA. LODs of the final method were reported as 2.9 × 10^3^ GE/5 g raw sausage and 5.3 × 10^4^ GE/2 g liver sausage. The method was applied to the analysis of liver sausages and raw pork sausages purchased at retail. MS2 bacteriophage was added to the food samples before processing. MS2 was detected in a separate RTPCR to HEV, thus functioning as both SPC and heterologous EAC. The method of Szabo et al. [48] was subsequently used in Swiss surveys that detected HEV in local salami-type sausages made from raw cured pig or game liver and meat [49,50]; the LOD was determined as being 1.56 × 10^3^ and 1.56 × 10^2^ per g of liver sausages and raw meat sausages, respectively [49]. The LOQ of the Szabo et al. [48] method was determined as 3.15 log GE/g [51], although this was performed using plasmid DNA and is therefore not likely to be accurate. The Szabo et al. [48] method was evaluated in a ring trial involving nine German and Swiss laboratories using artificially contaminated liver sausage as test material, and shown to be highly repeatable and reproducible. It was reported [52] that significantly higher recoveries of internal control swine mitochondrial sequences could be obtained with the method of Szabo et al. [48], by increasing the intensity and time of the bead-beating step. 

A relatively simple sample treatment procedure was used by Boxman et al. [53] in a method to detect HEV in porcine blood products used as food ingredients. Again, the assay of Jothikumar et al. [20] was used, with a standardised HEV RNA oligonucleotide [34] as EAC. Quantitation was performed with DNA standards. The method was subsequently used to detect HEV in pork liver, meat, and pate, and in wild boar meat samples [54].

Using a sample treatment based on vacuum filtration, Mykytczuk et al. [55] analysed pork products including pate, sausages, and liver for HEV. Following an in-house conventional PCR to screen for HEV-positive samples, digital droplet RTPCR (ddRTPCR) was performed using the primer/probe set of Jothikumar et al. [20] to quantify the viral load. The efficiency of detection was calculated using recovery of the SPC, and was found to be at least 1% in the majority of samples tested.

## 4. Conclusions

When discussing detection, we cannot fully ignore the implications of the extraction methods involved, which may affect the downstream results. Several sample treatment procedures have been developed for extraction of viruses prior to NA assay. The final steps of sample treatment have generally employed commercially available nucleic acid extraction kits, but various initial steps to prepare the sample so that it can be delivered in a form suitable for nucleic acid extraction have been described. Table 1 summarises the main features of initial sample treatment procedures developed for detection of HEV in pork products.

**Table 1 microorganisms-10-00428-t001:** Main features of sample treatment procedures prior to nucleic acid extraction.

Matrix	Sample Size (g)	Sample Treatment (Prior to NA Extraction)	Reference
liver	5–20 g	Blending in PBS	[41]
liver	150 mg	Homogenisation by scalpel, bead disruption, proteinase K	[32]
liver	0.1 mg	Homogenisation by beating with zirconia beads, lysis reagent, chloroform, centrifugation, gel separation	[56]
dried and liquid blood products	200 mg	Mixing with glycine buffer + beef extract	[53]
figatellu	10 mg	Fat discarded, homogenisation in PBS, centrifugation	[26]
liver, kidney, heart	1 cm^3^	As [32] then lysis reagent and chloroform extraction	[36]
liver, sausage, figatellu	3 g	Cell disruption in dH_2_O	[47]
liver, meat	10 mg	Bead disruption	[22]
liver sausage	3 g	Stomaching in dH_2_O, centrifugation	[44]
liver, pate, raw sausages		Homogenisation (ultrasonication?) in Glycine buffer pH9.5, filtration, centrifugation, PEG precipitation, lysis reagent	[55]
liver	1–10 g	As [32] then ultrafiltration	[39]
salami, boar liver	salami, 5 g; boar liver, 2 g	Stomaching in 7 mL lysis reagent centrifugation, chloroform extraction	[48]
liver	10–20 mg	Homogenisation by mortar and pestle	[27]
liver, chops	liver, 312 mg; chops, 262 mg	Mechanical disruption in lysis buffer, centrifugation	[23]

It is difficult to identify the best sample treatment process from the available information. A thorough evaluation of homogenisation procedures and extraction buffers is necessary, using a properly calibrated qRTPCR assay to compare extraction efficiencies. The most commonly used procedures appear to be based on either tissue grinding or cell disruption. Ideally, the extraction procedure should not rely on homogenisation apparatus or complex equipment; hazardous reagents should also be considered, if a standard procedure available for universal use is to be proposed. Sample treatments may require some variation in detail for different sample types, e.g., liver and meat. The variation in extraction methods highlights the importance of using appropriate controls in the detection methods to avoid any false negatives.

Various SPC viruses, e.g., FCV and MNoV, have been used in methods to detect HEV in pork products. Some commercial analyses are performed using mengovirus (MgV) (unpublished data). These viruses may not accurately reflect the characteristics of HEV and the effects of the extraction procedure on the virus. SPC viruses ideally should be as closely related to the target virus as possible [57]. To allow for production of stocks, the virus should be culturable, but there are very few HEV-related viruses which can be grown in vitro. Fish hepevirus is a member of the same family as HEV and can be cultured [58], so may be worth investigating as a candidate SPC. 

Few studies have reported extraction efficiencies or LOD/LOQs; some studies which have, have used an inappropriate assay calibration, making the data unreliable. Extraction efficiencies should be calculated taking into account recovery of SPC, and any inhibition of the RTPCR assay [9]. 

The RTPCR assay of Jothikumar et al. [20] appears to be the best assay for use as the basis of a standardised method for analysis of HEV in pork products. It has been used as the basis for several subsequent methods for HEV detection in pork products, and is widely used in laboratories performing HEV diagnostics [59]. The assay has been reported as being more sensitive than alternative RTPCRs [60], and is capable of detecting at least seven HEV genotypes including gts 1–4 [61]. It can be modified to incorporate a minor groove binder to increase the hybridization stability of the probe; this has been reported to reduce false-negative results [62]. An efficient RNA IAC exists for use in this assay [33], which could also be used for calibration. 

Although international standards exist for the detection of hepatitis A virus and norovirus in various food matrices [8,9], there is currently no standard method for detection of HEV in pork products. Meanwhile, evaluation and optimisation of key sample treatment procedures, and thorough performance characterisation with precise determination of extraction efficiencies and LODs/LOQs, will produce the ideal candidate HEV/pork product detection method for international standardisation.

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
