# Peer review of "Real-Time PCR-Based Methods for Detection of Hepatitis E Virus in Pork Products: A Critical Review"

_microorganisms, 2022, doi:10.3390/microorganisms10020428_

Round 1

Reviewer 1 Report

Though interesting and providing a large set of data, the presentation does not support the publication of the paper as it is.

First of all, it is not indicated how is performed the review. According to Microorganism’s instructions for authors for Reviews “Systematic reviews should follow the PRISMA guidelines”. The manuscript is lacking of all the methodological part of the study.

The introduction is lacking of the aim of the manuscript. Line 54, methods based on cell culture not are for detection (as stated by authors) but for determination of HEV infectivity. This statement could confuse the readers.

Part 2 is not well written. The large use of acronyms and abbreviation (eg. qPCR, RTPCR, RTqPCR, qRTPCR), often incorrectly, does not make easy to understand the text.

Part3 is focused mainly on pre-analytical treatment and the molecular methods are only listed. It is suggested to create a new subsection in the manuscript to discuss the sample treatment procedures. Moreover, there is confusion among references stated in the text (eg, line 165; line 192 Di Bartolo not Di Bartollo). What are you referring to the abbreviation GE? (line 185, 206, 244).

The assay of Jothikumar et al. (2006) [please correct the year in the references section] was optimized by the inclusion of a modified probe for better discrimination between negative samples and those containing only low amounts of RNA (Garson et al., 2012) [not cited in the text].

Other important studies for use to define a standardized method for HEV in pork products, or in general in food, are not included in the discussion (La Rosa et al., 2017; Di Pasquale et al., 2019).

Author Response

Though interesting and providing a large set of data, the presentation does not support the publication of the paper as it is.

First of all, it is not indicated how is performed the review. According to Microorganism’s instructions for authors for Reviews “Systematic reviews should follow the PRISMA guidelines”. The manuscript is lacking of all the methodological part of the study.

Authors’ Response. This is not a systematic review; it is simply a critical one. We have amended the title of the paper to reflect this.

The introduction is lacking of the aim of the manuscript. Line 54, methods based on cell culture not are for detection (as stated by authors) but for determination of HEV infectivity. This statement could confuse the readers.

Authors’ Response. There is evidence in the literature where detection of HEV by IFA in cell culture has been used; we have amended the sentence to clarify. Thank you for pointing this out, the aim has been reiterated in the introduction.

Part 2 is not well written. The large use of acronyms and abbreviation (eg. qPCR, RTPCR, RTqPCR, qRTPCR), often incorrectly, does not make easy to understand the text.

Authors’ Response. We have checked the text regarding the acronyms for consistency and have amended where required. Section 2 fully defines what the acronyms stand for, especially in lines 93-95. This is the appropriate nomenclature for these methods. Section 2 is relevant to define the aspects which are important in the reported studies and to explain why controls are important and not often included in the literature.

Part3 is focused mainly on pre-analytical treatment and the molecular methods are only listed. It is suggested to create a new subsection in the manuscript to discuss the sample treatment procedures. Moreover, there is confusion among references stated in the text (eg, line 165; line 192 Di Bartolo not Di Bartollo). What are you referring to the abbreviation GE? (line 185, 206, 244).

Authors’ Response. We have amended the section to remove references to specific extraction methods as the title is specifically about PCR detection. Extraction details are summarised in table 1 and additions have been made.  The spelling error and the citation have now been corrected. The term “genome equivalents” (as used in several studies) has now been clarified.

The assay of Jothikumar et al. (2006) [please correct the year in the references section] was optimized by the inclusion of a modified probe for better discrimination between negative samples and those containing only low amounts of RNA (Garson et al., 2012) [not cited in the text].

Authors’ Response. The error in the Jothikumar reference has been now corrected. The Garson et al reference has now been added.

Other important studies for use to define a standardized method for HEV in pork products, or in general in food, are not included in the discussion (La Rosa et al., 2017; Di Pasquale et al., 2019).

Authors’ Response. La Rosa (2017) can not be identified. If La Rosa (2018) is meant, that study focusses on shellfish and seawater and is therefore not appropriate for this review. The paper of Di Pasquale et al. (2019) details a survey of HEV in wild boar. Previously published methods were used, and no new or modified methodology, or new information regarding e.g. LOD or LOQ, was reported. It is not considered that the inclusion of this reference would add to the review.

Reviewer 2 Report

Dear Editor-in-Chief

Thanks for the attention and guidance.

The article "Real-time PCR-based methods for detection of hepatitis E virus in pork products: a review" is excellent and desirable in terms of scientific structure. The article is in line with the goals of the journal. The introduction contains enough information about the subject of the study. The methodology is accurate, precise, and well presented. Results are original, well presented, and contribute to the advancement of understanding detection of hepatitis E virus in food of an animal origin. The discussion is very well written and sufficiently discusses the obtained results. Given my experience in the field, I concur that RTPCR assay by Jothikumar et al. at this moment seems to be the best candidate for the standardized method of HEV detection in food of an animal origin.

After careful cross-check of sample treatment procedures described in Table 1 (Page 6/11, Line 304) and available literature data, I would recommend the corresponding author to include one more article in this section (Milojević et al., 2019. Screening and Molecular Characterization of Hepatitis E Virus in Slaughter Pigs in Serbia. Food and environmental virology, 11(4), 410–419. https://doi.org/10.1007/s12560-019-09393-1) since this paper described liver treatment with zirconia beads, Trizol-chloroform, and phase-lock gel which was different than other treatments described in the aforementioned table.

Author Response

After careful cross-check of sample treatment procedures described in Table 1 (Page 6/11, Line 304) and available literature data, I would recommend the corresponding author to include one more article in this section (Milojević et al., 2019. Screening and Molecular Characterization of Hepatitis E Virus in Slaughter Pigs in Serbia. Food and environmental virology, 11(4), 410–419. https://doi.org/10.1007/s12560-019-09393-1) since this paper described liver treatment with zirconia beads, Trizol-chloroform, and phase-lock gel which was different than other treatments described in the aforementioned table.

Authors’ Response. We thank the reviewer for the suggestion and this has been added to Table 1.

Reviewer 3 Report

The main purpose of this review paper is to address the detection method for hepatitis E virus (HEV) in pork products. The review centers around four main sections: an introduction outlining the major background about the detection method of Hepatitis related pathogens in pork-based product; A detailed overview of the reverse transcription polymerase chain reaction (RT-PCR)-based detection of the foodborne virus; A detailed overview on Real-time RT-PCR-based detection methods for HEV in pork products; and remarks on the immediacy and importance about the standard method for detection of HEV in pork products.

Overall, the manuscript is written in an excellent way, with interesting information for readers involved in the published HEV detection methods, with emphasis on the detection of HEV in pork products and the similar foodstuffs. The summary table 1 (in section 4) is excellent illustration of the main features of initial sample treatment procedures developed for HEV antigen detection in pork products.

Major comments-

However, the manuscript in section 3 seems a bit dispersed. The author just narrated the content of the article one by one, the text jumps from one study to another without the necessary summary about the results. In addition, some of the referenced papers in this section are not about the HEV detection method, which is not related to the main topic of subtitle 3. Thus, improvements in the text are required for flow and to improve the reader's comprehension. In addition, references not related to the subtitle should be removed and new relevant ones should be added. A summary table should be included. I recommend the authors should consider major revision of the text in this part.   

Minor comments-

  1. Line 52, please add a “.” In the last word.
  2. Line 61, please delete the redundant “.” after “HEV”.
  3. Line 304, the sample size should be 5-20 g.

Author Response

Major comments-

However, the manuscript in section 3 seems a bit dispersed. The author just narrated the content of the article one by one, the text jumps from one study to another without the necessary summary about the results. In addition, some of the referenced papers in this section are not about the HEV detection method, which is not related to the main topic of subtitle 3. Thus, improvements in the text are required for flow and to improve the reader's comprehension. In addition, references not related to the subtitle should be removed and new relevant ones should be added. A summary table should be included. I recommend the authors should consider major revision of the text in this part.   

Authors’ Response: We thank the reviewer for their comment as it is important for the manuscript to flow well. The text has been edited as tracked changes and as a clean copy which we believe improves understanding and clarity. It is noteworthy that the literature was searched for content on PCR methods that may not be reflected in the title of the article cited.

Minor comments-

1. Line 52, please add a “.” In the last word.

Authors’ Response: Done.

2. Line 61, please delete the redundant “.” after “HEV”.

Authors’ Response: Done.

3. Line 304, the sample size should be 5-20 g.

Authors’ Response: Done.

Round 2

Reviewer 1 Report

Despite the changesI do not believe that the manuscript is suitable for publication in an important journal as Microorganisms.

According to this reviewer, it would have been more interesting if the authors decided to focus the review only on the diverse extraction methods.

To reply to the authors: The study by Di Pasquale et al. (2019), although used a modified protocol of Szabo et al. (2015) for virus extraction and the molecular method of Jothikumar et al. (2016), provided an important starting point to the definition of a standard method for the detection of HEV in food. It defined a quantitative method with a sample process control, an amplification control and a real-time RT-(q)PCR assay with a LOD95 egual to 1.12 genome copies/μl ≡ 2.6 × 102 g.c./g. A such study cannot be neglected.

The work isn't well organized (e.g., the numeration of the references in the text is wrong).

It does not add any additional information than that stated in the scientific opinion of the EFSA (https://doi.org/10.2903/j.efsa.2017.4886).

Reviewer 3 Report

The authors made extensive improvements at this version, it should be published in its current state.